# DEEP SYMBOLIC REGRESSION

## ABSTRACT

Discovering the underlying mathematical expressions describing a dataset is a core challenge for artificial intelligence. This is the problem of *symbolic regression*. Despite recent advances in training neural networks to solve complex tasks, deep learning approaches to symbolic regression are lacking. We propose a framework that combines deep learning with symbolic regression via a simple idea: use a large model to search the space of small models. More specifically, we use a recurrent neural network to emit a distribution over tractable mathematical expressions, and employ reinforcement learning to train the network to generate better-fitting expressions. Our algorithm significantly outperforms standard genetic programming-based symbolic regression in its ability to exactly recover symbolic expressions on a series of benchmark problems, both with and without added noise. More broadly, our contributions include a framework that can be applied to optimize hierarchical, variable-length objects under a black-box performance metric, with the ability to incorporate a priori constraints in situ.

## 1 INTRODUCTION

Understanding the mathematical relationships among variables in a physical system is an integral component of the scientific process. Symbolic regression aims to identify these relationships by searching over the space of tractable mathematical expressions to best fit a dataset. Specifically, given a dataset of $(X, y)$ pairs, where $X \in \mathbb{R}^n$ and $y \in \mathbb{R}$, symbolic regression aims to identify a function $f(X) : \mathbb{R}^n \to \mathbb{R}$ that minimizes a distance metric $D(y, f(X))$ between real and predicted values. That is, symbolic regression seeks to find the optimal $f^\star = \arg\min_f D(y, f(X))$, where the functional form of $f$ is a tractable expression.

The resulting expression $f^\star$ may be readily interpretable and/or provide useful scientific insights simply by inspection. In contrast, conventional regression imposes a single model structure that is fixed during training, often chosen to be expressive (e.g. a neural network) at the expense of being easily interpretable. However, the space of mathematical expressions is discrete (in model structure) and continuous (in model parameters), growing exponentially with the length of the expression, rendering symbolic regression an extremely challenging machine learning problem.

Given the large and combinatorial search space, traditional approaches to symbolic regression typically utilize evolutionary algorithms, especially genetic programming (GP) (Koza, 1992; Bäck et al., 2018). In GP-based symbolic regression, a population of mathematical expressions is "evolved" using evolutionary operations like selection, crossover, and mutation to improve a fitness function. While GP can be effective, it is also known to scale poorly to larger problems and to exhibit high sensitivity to hyperparameters.

Deep learning has permeated almost all areas of artificial intelligence, from computer vision (Krizhevsky et al., 2012) to optimal control (Mnih et al., 2015). However, deep learning may seem incongruous with or even antithetical toward symbolic regression, given that neural networks are typically highly complex, difficult to interpret, and rely on gradient information. We propose a framework that resolves this incongruity by tying deep learning and symbolic regression together with a simple idea: use a large model (i.e. neural

network) to search the space of small models (i.e. symbolic expressions). This framework leverages the representational capacity of neural networks while entirely bypassing the need to interpret a network.

We present deep symbolic regression (DSR), a gradient-based approach for symbolic regression based on reinforcement learning. In DSR, a recurrent neural network (RNN) emits a distribution over mathematical expressions. Expressions are sampled from the distribution, instantiated, and evaluated based on their fitness to the dataset. This fitness is used as the reward signal to train the RNN parameters using a policy gradient algorithm. As training proceeds, the RNN adjusts the likelihood of an expression relative to its reward, assigning higher probabilities to better fitting expressions.

We demonstrate that DSR outperforms a standard GP implementation in its ability to recover exact symbolic expressions from data, both with and without added noise. We summarize our contributions as follows: 1) a novel method for solving symbolic regression that outperforms standard GP, 2) an autoregressive generative modeling framework for optimizing hierarchical, variable-length objects, 3) a framework that accommodates in situ constraints, and 4) a novel risk-seeking strategy that optimizes for best-case performance.

## 2 RELATED WORK

**Symbolic regression.** Symbolic regression has a long history of evolutionary strategies, especially GP (Koza, 1992; Bäck et al., 2018; Uy et al., 2011). Among non-evolutionary approaches, the recent AI Feynman algorithm (Udrescu & Tegmark, 2019) is a multi-staged approach to symbolic regression leveraging the observation that physical equations often exhibit simplifying properties like multiplicative separability and translational symmetry. The algorithm identifies and exploits such properties to recursively define simplified sub-problems that can eventually be solved using simple techniques like a polynomial fit or small brute force search. Brunton et al. (2016) develop a sparse regression approach to recover nonlinear dynamics equations from data; however, their search space is limited to linear combinations of a library of basis functions.

**AutoML.** Our framework has many parallels to a body of works within automated machine learning (AutoML) that use an autoregressive RNN to define a distribution over discrete objects and use reinforcement learning to optimize this distribution under a black-box performance metric (Zoph & Le, 2017; Ramachandran et al., 2017; Bello et al., 2017). The key methodological difference to our framework is that these works optimize objects that are both *sequential* and *fixed length*. For example, in neural architecture search (Zoph & Le, 2017), an RNN searches the space of neural network architectures, which are encoded by a sequence of discrete "tokens" specifying architectural properties (e.g. number of neurons) of each layer. The length of the sequence is fixed or scheduled during training. In contrast, a major contribution of our framework is defining a search space that is both inherently hierarchical and variable length.

The most similar AutoML work searches for neural network activation functions (Ramachandran et al., 2017). While the space of activation functions is hierarchical in nature, the authors (rightfully) constrain this space substantially by positing a functional unit that is repeated sequentially, thus restricting their search space back to a fixed-length sequence. This constraint is well-justified for learning activation functions, which tend to exhibit similar hierarchical structures. However, a repeating-unit constraint is not practical for symbolic regression because the ground truth expression may have arbitrary structure.

**Autoregressive models.** The RNN-based distribution over expressions used in DSR is autoregressive, meaning each token is conditioned on the previously sampled tokens. Autoregressive models have proven to be useful for audio and image data (Oord et al., 2016a;b) in addition to the AutoML works discussed above; we further demonstrate their efficacy for hierarchical expressions.

GraphRNN defines a distribution over graphs that generates an adjacency matrix one column at a time in autoregressive fashion (You et al., 2018). In principle, we could have constrained GraphRNN to define the distribution over expressions, since trees are a special case of graphs. However, GraphRNN constructs

graphs breadth-first, whereas expressions are more naturally represented using depth-first traversals (Li et al., 2005). Further, DSR exploits the hierarchical nature of trees by providing the parent and sibling as inputs to the RNN, and leverages the additional structure of expression trees that a node's value determines its number of children (e.g. cosine is a unary node).

## 3 METHODS

Our overall approach involves representing mathematical expressions by the pre-order traversals of their corresponding symbolic expression trees, developing an autoregressive model to generate expression trees under a pre-specified set of constraints, and using reinforcement learning to train the model to generate better-fitting expressions.

### 3.1 GENERATING EXPRESSIONS WITH A RECURRENT NEURAL NETWORK

We leverage the fact that algebraic expressions can be represented using symbolic expression trees, a type of binary tree in which nodes map to mathematical operators, input variables, or constants. Operators are internal nodes and may be unary (e.g. sine) or binary (e.g. multiply). Input variables and constants are terminal nodes. We encode an expression $\tau$ by the pre-order traversal (i.e. depth-first, then left-to-right) of its corresponding expression tree.[1] We denote the $i^{\text{th}}$ node in the traversal as $\tau_i$ and the length of the traversal as $|\tau| = T$. Each node has a value within a given library $\mathcal{L}$ of possible node values or "tokens," e.g. $\{+, -, \times, \div, \sin, \cos, x\}$.

Expressions are generated one node at a time along the pre-order traversal (from $\tau_1$ to $\tau_T$). For each node, a categorical distribution with parameters $\psi$ defines the probabilities of selecting each node value from $\mathcal{L}$. To capture the "context" of the expression as it is being generated, we condition this probability upon the selections of all previous nodes in that traversal. This conditional dependence can be achieved very generally using an RNN with parameters $\theta$ that outputs a probability vector $\psi$ in autoregressive manner.

Specifically, the $i^{\text{th}}$ output vector $\psi^{(i)}$ of the RNN defines the probability distribution for selecting the $i^{\text{th}}$ node value $\tau_i$, conditioned on the previously selected node values $\tau_{1:(i-1)}$:

$$p(\tau_i|\tau_{1:(i-1)}; \theta) = \psi^{(i)}_{\mathcal{L}(\tau_i)},$$

where $\mathcal{L}(\tau_i)$ is the index in $\mathcal{L}$ corresponding to node value $\tau_i$. The likelihood of the sampled expression is computed using the chain rule of conditional probability:

$$p(\tau|\theta) = \prod_{i=1}^{|\tau|} p(\tau_i|\tau_{1:(i-1)}; \theta) = \prod_{i=1}^{|\tau|} \psi^{(i)}_{\mathcal{L}(\tau_i)}$$

The sampling process is illustrated in Figure 1 and described in Algorithm 1. Additional algorithmic details of the sampling process are described in Subroutines 1 and 2 in Appendix A. Starting at the root node, a node value is sampled according to $\psi^{(1)}$. Subsequent node values are sampled autoregressively in a depth-first, left-to-right manner until the tree is complete (i.e. all tree branches reach terminal nodes). The resulting sequence of node values is the tree's pre-order traversal, which can be used to reconstruct the tree[2] and its

---

[1]Given an expression tree (or equivalently, its pre-order traversal), the corresponding mathematical expression is unique; however, given an expression, its expression tree (or its corresponding traversal) is not unique. For example, $x^2$ and $x \cdot x$ are equivalent expressions but yield different trees. For simplicity, we use $\tau$ somewhat abusively to refer to an expression where it technically refers to an expression tree (or equivalently, its corresponding traversal).

[2]In general, a pre-order traversal is insufficient to uniquely reconstruct the tree. However, in this context, we know how many child nodes each node has based on its value, e.g. "multiply" is a binary operator and thus has two children.

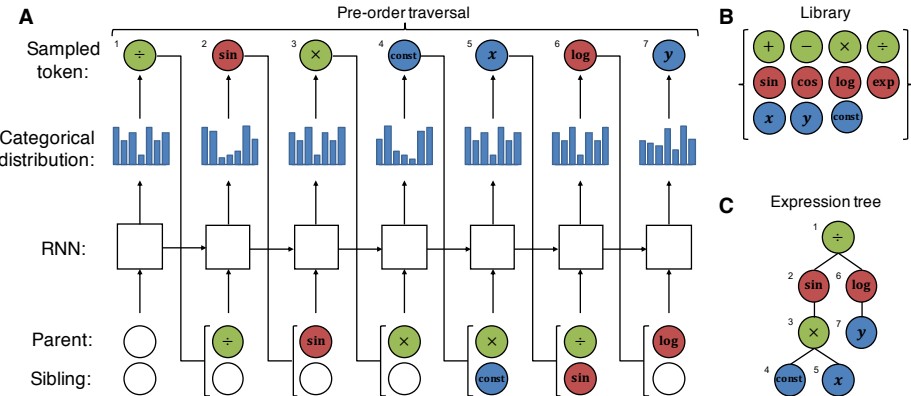

Figure 1: **A**. Sampling an expression from the RNN. Nodes are selected one at a time in autoregressive fashion along the pre-order traversal of the corresponding expression tree. For each token, the RNN outputs a categorical distribution over tokens, a token is sampled, and the parent and sibling of the next token are used as the next input to the RNN. In this example, the sampled expression is $\sin(cx)/\log(y)$, where the value of the constant $c$ is optimized with respect to an input dataset. Numbers indicate the order in which tokens were sampled. Colors correspond to the arity of the token. White circles represent empty tokens. **B**. The library of tokens. **C**. The expression tree sampled in **A**.

corresponding expression. Note that different samples of the distribution have different tree structures of different size. Thus, the search space is inherently both hierarchical and variable length.

**Providing hierarchical inputs to the RNN.** Naively, the input to the RNN when sampling $\tau_i$ would be a representation (i.e. embedding or one-hot encoding) of the previously sampled token, $\tau_{i-1}$. Indeed, this is typical in related autoregressive models, e.g. when generating sentences (Vaswani et al., 2017) or for neural architecture search (Zoph & Le, 2017). However, the search space for symbolic regression is inherently hierarchical, and the previously sampled token may actually be very distant from the next token to be sampled in the expression tree. For example, the fifth and sixth tokens sampled in Figure 1 are adjacent nodes in the traversal but are four edges apart in the expression tree. To better capture hierarchical information, we provide as inputs to the RNN a representation of the parent and sibling node of the token being sampled. We introduce an empty token for cases in which a node does not have a parent or sibling. Pseudocode for identifying the parent and sibling nodes given a partial traversal is provided in Subroutine 2 in Appendix A.

**Constraining the search space.** Under our framework, it is straightforward to apply a priori constraints to reduce the search space. To demonstrate, we impose several simple, domain-agnostic constraints: (1) Expressions are limited to a pre-specified minimum and maximum length. We selected minimum length of 2 to prevent trivial expressions and a maximum length of 30 to ensure expressions are tractable. (2) The children of an operator should not all be constants, as the result would simply be a different constant. (3) The child of a unary operator should not be the inverse of that operator, e.g. $\log(\exp(x))$ is not allowed. (4) Direct descendants of trigonometric operators should not be trigonometric operators, e.g. $\sin(x + \cos(x))$ is not allowed because cosine is a descendant of sine. While still semantically meaningful, such composed trigonometric operators do not appear in virtually any scientific domain.

We apply these constraints in situ (concurrently with autoregressive sampling) by zeroing out the probabilities of selecting tokens that would violate a constraint. Pseudocode for this process is provided in Subroutine

---

For domains without this property, the number of children can be sampled from an additional RNN output. A pre-order traversal plus the corresponding number of children for each node is sufficient to uniquely reconstruct the tree.

---

**Algorithm 1:** Sampling an expression from the RNN

---

1   **function** SampleExpression($\theta, \mathcal{L}$)

     **input** : RNN with parameters $\theta$; library of tokens $\mathcal{L}$

     **output:** Pre-order traversal $\tau$ of an expression sampled from the RNN

2      $\tau = []$                               `// Empty list`

3      $x = \text{empty} \| \text{empty}$        `// Initial RNN input is empty parent and sibling`

4      $h_0 = \vec{0}$                      `// Initialize RNN cell state to zero`

5      **for** $i = 1, \ldots, T$ **do**

6          $(\psi^{(i)}, h_i) = \text{RNN}(x, h_{i-1}; \theta)$

7          $\psi^{(i)} \leftarrow \text{ApplyConstraints}(\psi^{(i)}, \mathcal{L}, \tau)$       `// Adjust probabilities`

8          $\tau_i = \text{Categorical}(\psi^{(i)})$           `// Sample next token`

9          $\tau \leftarrow \tau \| \tau_i$          `// Append token to traversal`

10         **if** ExpressionComplete($\tau$) **then**

11             **return** $\tau$

12         $x \leftarrow \text{ParentSibling}(\tau)$      `// Compute next parent and sibling`

13      **end**

14      **return** $\tau$

---

**Algorithm 2:** Deep symbolic regression

---

1   **function** DSR($\alpha, N, \mathcal{L}, X, y$)

     **input** : learning rate $\alpha$; batch size $N$; library of tokens $\mathcal{L}$; input dataset $(X, y)$

     **output:** Best fitting expression $\tau^{\star}$

2      Initialize RNN with parameters $\theta$, defining distribution over expressions $p(\cdot|\theta)$

3      $\tau^{\star} = \text{null}$

4      $b = 0$

5      **repeat**

6          $\mathcal{T} = \{\tau^{(i)} \sim p(\cdot|\theta)\}_{i=1:N}$        `// Sample expressions (Algorithm 1)`

7          $\mathcal{T} \leftarrow \{\text{OptimizeConstants}(\tau^{(i)}, X, y)\}_{i=1:N}$      `// Optimize constants`

8          $\mathcal{R} = \{R(\tau^{(i)}) - \lambda_{\mathcal{C}} \mathcal{C}(\tau^{(i)})\}_{i=1:N}$      `// Compute rewards`

9          $\hat{g} = \frac{1}{N} \sum_{i=1}^{N} R(\tau^{(i)}) \nabla_{\theta} \log p(\tau^{(i)}|\theta)$      `// Compute policy gradient`

10        $\theta \leftarrow \theta + \alpha(\hat{g_1} + \hat{g_2})$             `// Apply gradients`

11        **if** $\max \mathcal{R} > R(\tau^{\star})$ **then** $\tau^{\star} \leftarrow \tau^{(\arg\max \mathcal{R})}$    `// Update best expression`

12      **return** $\tau^{\star}$

---

1 in Appendix A. This process ensures that all samples adhere to all constraints, without rejecting samples post hoc. In contrast, imposing constraints in GP-based symbolic regression can be problematic (Craenen et al., 2001). In practice, evolutionary operations that violate constraints are typically rejected post hoc (Fortin et al., 2012).

### 3.2 TRAINING THE RNN USING POLICY GRADIENTS

**Optimizing the parameters of the sampled expressions.** Once a pre-order traversal is sampled, we instantiate the corresponding symbolic expression. The expression may have several constant tokens, which can be viewed as model parameters. We train these model parameters by minimizing the mean-squared error

with respect to an input dataset using a nonlinear optimization algorithm, e.g. BFGS (Fletcher, 2013). We perform this inner optimization loop for each sampled expression before training the RNN.

**Training the RNN using policy gradients.** Given a distribution over mathematical expressions $p(\tau|\theta)$ and a measure of performance of an expression $R(\tau)$, we consider the objective to maximize $J(\theta)$, defined as the expectation of $R$ under expressions sampled from the distribution:

$$J(\theta) \equiv \mathbb{E}_{\tau \sim p(\tau|\theta)} \left[ R(\tau) \right]$$

We use REINFORCE (Williams, 1992) to maximize this expectation via gradient ascent:

$$\begin{aligned} \nabla_\theta J(\theta) &= \nabla_\theta \mathbb{E}_{\tau \sim p(\tau|\theta)} \left[ R(\tau) \right] \\ &= \mathbb{E}_{\tau \sim p(\tau|\theta)} \left[ R(\tau) \nabla_\theta \log p(\tau|\theta) \right] \end{aligned}$$

This result allows us to estimate the expectation using samples from the distribution. Specifically, we can obtain an unbiased estimate of $\nabla_\theta J(\theta)$ by computing the sample mean over a batch of $N$ sampled expressions $\mathcal{T} = \{\tau^{(i)}\}_{i=1:N}$:

$$\nabla_\theta J(\theta) \approx \frac{1}{N} \sum_{i=1}^{N} R(\tau^{(i)}) \nabla_\theta \log p(\tau^{(i)}|\theta)$$

**Reward function.** A standard fitness measure in GP-based symbolic regression is normalized root-mean-square error (NRMSE), the root-mean-square error normalized by the standard deviation of the target values, $\sigma_y$. That is, given a dataset of $n$ number of $(X, y)$ pairs, NRMSE $= \frac{1}{\sigma_y} \sqrt{\frac{1}{n} \sum_{i=1}^{n} (y_i - \hat{y}_i)^2}$, where $\hat{y} = f(X)$ are the predicted values computed using the candidate expression $f$. Normalization by $\sigma_y$ makes the metric commensurate across different datasets with potentially different ranges. However, metrics based on mean-square error exhibit extraordinarily large values for some expressions, e.g. an expression that incorrectly divides by an input variable with values near zero. For a gradient-based approach like DSR, this results in the gradient being dominated by the worst expressions, which can lead to instability. We found that a bounded reward function is more stable; thus, we applied a squashing function, yielding the reward function $R(\tau) = 1/(1 + \text{NRMSE})$.[3]

We introduce the "vanilla" version of DSR in Algorithm 2. Below we describe several simple extensions.

**Reward baseline.** The above approximation to $\nabla_\theta J(\theta)$ is an unbiased gradient estimate, but in practice has high variance. To reduce variance, we include a baseline function $b$:

$$\nabla_\theta J(\theta) \approx \frac{1}{N} \sum_{i=1}^{N} \left[ R(\tau^{(i)}) - b \right] \nabla_\theta \log p(\tau^{(i)}|\theta)$$

As long as the baseline is not a function of the current batch of expressions, the gradient estimate is still unbiased. We define the baseline function as an exponentially-weighted moving average of batches of rewards. Intuitively, the gradient step increases the likelihood of expressions above the baseline and decreases the likelihood of expressions below the baseline.

**Complexity penalty.** We include an optional complexity penalty that is added to the reward function. For simplicity, we consider the complexity metric $|\tau|$, i.e. the number of nodes in the expression tree. More complicated metrics have been proposed that capture hierarchical features of the tree and/or deduced properties of the resulting expression (Vladislavleva et al., 2008).

---

[3]Since GP-based approaches using tournament selection only rely on the *rankings* of the fitness measure within the population, large fitness values are not problematic. Since $R(\tau)$ is monotonic in NRMSE, GP is unaffected by squashing.

---

**Algorithm 3:** Deep symbolic regression with baseline, risk-seeking, entropy bonus, and complexity penalty

---

1  **function** DSR($\alpha, \beta, \lambda_{\mathcal{C}}, \lambda_{\mathcal{H}}, \epsilon, N, \mathcal{L}, X, y$)

    **input** : learning rate $\alpha$; moving average coefficient $\beta$; complexity coefficient $\lambda_{\mathcal{C}}$; entropy coefficient $\lambda_{\mathcal{H}}$; risk factor $\epsilon$; batch size $N$; library of tokens $\mathcal{L}$; input dataset $(X, y)$

    **output:** Best fitting expression $\tau^{\star}$

2     Initialize RNN with parameters $\theta$, defining distribution over expressions $p(\cdot|\theta)$

3     $\tau^{\star}$ = null

4     $b = 0$

5     **repeat**

6         $\mathcal{T} = \{\tau^{(i)} \sim p(\cdot|\theta)\}_{i=1:N}$       // Sample expressions (Algorithm 1)

7         $\mathcal{T} \leftarrow \{\text{OptimizeConstants}(\tau^{(i)}, X, y)\}_{i=1:N}$     // Optimize constants

8         $\mathcal{R} = \{R(\tau^{(i)}) - \lambda_{\mathcal{C}}\mathcal{C}(\tau^{(i)})\}_{i=1:N}$     // Compute rewards

9         $R_{\epsilon} = (1 - \epsilon)$ percentile of $\mathcal{R}$     // Compute threshold

10        $\mathcal{T} \leftarrow \{\tau^{(i)} : R(\tau^{(i)}) \geq R_{\epsilon}\}$     // Select subset of expressions

11        $\mathcal{R} \leftarrow \{R(\tau^{(i)}) : R(\tau^{(i)}) \geq R_{\epsilon}\}$     // Select subset of rewards

12        $\hat{g_1} = \text{ReduceMean}((\mathcal{R} - b)\nabla_{\theta}\log p(\mathcal{T}|\theta))$     // Compute policy gradient

13        $\hat{g_2} = \text{ReduceMean}(\lambda_{\mathcal{H}}\nabla_{\theta}\mathcal{H}(\mathcal{T}|\theta))$     // Compute entropy gradient

14        $\theta \leftarrow \theta + \alpha(\hat{g_1} + \hat{g_2})$     // Apply gradients

15        $b \leftarrow \beta \cdot \text{ReduceMean}(\mathcal{R}) + (1 - \beta)b$     // Update baseline

16        **if** $\max \mathcal{R} > R(\tau^{\star})$ **then** $\tau^{\star} \leftarrow \tau^{(\arg\max \mathcal{R})}$     // Update best expression

17     **return** $\tau^{\star}$

---

**Entropy bonus.** We provide a bonus to the loss function proportional to the entropy of the sampled expressions. In accordance with the maximum entropy reinforcement learning framework (Haarnoja et al., 2018), this bonus serves two purposes. First, it encourages the RNN to explore more expressions, preventing premature convergence to a local optimum. In practice, this often leads to a better end result. Second, it encourages the RNN to assign equal likelihood to different expressions that have equal fitness.

**Risk-seeking.** The policy performance, $J$, is defined as an *expectation*. However, in practice, the performance of symbolic regression is measured by the single or few best expressions. Thus, we employ a novel risk-seeking technique in which only the top $\epsilon$ percentile samples from each batch are used in the gradient computation. This has the effect of increasing best-case performance at the expense of lower worst-case and average performances. This process is essentially the opposite of the EpOpt technique (Rajeswaran et al., 2016) used for risk-averse reinforcement learning, in which only the *bottom* $\epsilon$ percentile samples from each batch are used.

The complete algorithm, including reward baseline, complexity penalty, entropy bonus, and risk-seeking, is shown in Algorithm 3.

## 4   RESULTS AND DISCUSSION

**Evaluating DSR.** We evaluated DSR on a set of 12 commonly used symbolic regression benchmarks (Uy et al., 2011), as well as 4 additional variants in which we introduced real-valued constants to demonstrate the inner optimization loop. Each benchmark is defined by a ground truth expression, a training and testing dataset, and set of allowed operators, described in Table 2 in Appendix A. The training data is used to compute the reward for each candidate expression, the test data is used to evaluate the best found candidate

Table 1: Performance comparison of DSR and GP-based symbolic regression on 16 symbolic regression benchmarks. Bold values represent statistical significance (two-sample $t$-test, $p < 0.05$). Errors represent standard deviation ($n = 100$ for Nguyen benchmarks; $n = 10$ for Constant benchmarks).

| Benchmark | Expression | GP Recovery | GP NRMSE | DSR Recovery | DSR NRMSE |
|---|---|---|---|---|---|
| Nguyen-1 | $x^3 + x^2 + x$ | 51% | $0.009 \pm 0.012$ | **99%** | $\mathbf{0.000 \pm 0.000}$ |
| Nguyen-2 | $x^4 + x^3 + x^2 + x$ | 22% | $0.021 \pm 0.016$ | **100%** | $\mathbf{0.000 \pm 0.000}$ |
| Nguyen-3 | $x^5 + x^4 + x^3 + x^2 + x$ | 6% | $0.025 \pm 0.025$ | **100%** | $\mathbf{0.000 \pm 0.000}$ |
| Nguyen-4 | $x^6 + x^5 + x^4 + x^3 + x^2 + x$ | 3% | $0.024 \pm 0.012$ | **100%** | $\mathbf{0.000 \pm 0.000}$ |
| Nguyen-5 | $\sin(x^2)\cos(x) - 1$ | 20% | $0.048 \pm 0.154$ | 17% | $0.031 \pm 0.042$ |
| Nguyen-6 | $\sin(x) + \sin(x + x^2)$ | 29% | $0.019 \pm 0.024$ | **100%** | $\mathbf{0.000 \pm 0.000}$ |
| Nguyen-7 | $\log(x + 1) + \log(x^2 + 1)$ | 0% | $0.017 \pm 0.021$ | 1% | $0.020 \pm 0.021$ |
| Nguyen-8 | $\sqrt{x}$ | 1% | $0.063 \pm 0.032$ | **34%** | $0.081 \pm 0.112$ |
| Nguyen-9 | $\sin(x) + \sin(y^2)$ | 100% | $0.000 \pm 0.000$ | 100% | $0.000 \pm 0.000$ |
| Nguyen-10 | $2\sin(x)\cos(y)$ | 61% | $0.016 \pm 0.028$ | **100%** | $\mathbf{0.000 \pm 0.000}$ |
| Nguyen-11 | $x^y$ | 5% | $0.272 \pm 0.196$ | **100%** | $\mathbf{0.000 \pm 0.000}$ |
| Nguyen-12 | $x^4 - x^3 + \frac{1}{2}y^2 - y$ | 0% | $\mathbf{0.202 \pm 0.101}$ | 0% | $0.301 \pm 0.077$ |
| | Average | 24.8% | $0.060 \pm 0.023$ | **71.0%** | $\mathbf{0.036 \pm 0.012}$ |
| | | | | | |
| Constant-1 | $3.39x^3 + 2.12x^2 + 1.78x$ | 100% | $0.000 \pm 0.000$ | 100% | $0.000 \pm 0.000$ |
| Constant-2 | $\sin(x^2)\cos(x) - 0.75$ | 60% | $0.000 \pm 0.001$ | **100%** | $0.000 \pm 0.000$ |
| Constant-3 | $\sin(1.5x)\cos(0.5y)$ | 20% | $0.001 \pm 0.002$ | **90%** | $0.001 \pm 0.002$ |
| Constant-4 | $2.7x^y$ | 0% | $0.184 \pm 0.038$ | **80%** | $\mathbf{0.034 \pm 0.078}$ |
| | Average | 45.0% | $0.046 \pm 0.009$ | **92.5%** | $\mathbf{0.009 \pm 0.020}$ |

expression at the end of training, and the ground truth function is used to determine whether the best found candidate expression was correctly recovered.

As a baseline, we compared against standard GP-based symbolic regression. To ensure fair comparison, the same constant optimizer (BFGS) was used for both methods. We ran independent training runs for GP and DSR for each benchmark expression ($n = 100$ for benchmarks without constants; $n = 10$ for benchmarks with constants). For each experiment, we generated 1,000 candidate expressions per generation/iteration for 1,000 generations/iterations, resulting in 1,000,000 total expressions. For each training run, the expression with the best reward is selected and we record the NRMSE on the test data.

For GP, we used the open-source software package "deap" (Fortin et al., 2012). For DSR, the RNN comprised a single-layer LSTM of 32 hidden units. Additional hyperparameters and experiment details are provided in Appendix A.

In Table 1, we report the percentage of runs that correctly recover the expression and NRMSE on the test data for each benchmark. DSR significantly outperforms GP in its ability to exactly recover benchmark expressions. DSR also outperforms GP in the average NRMSE across all expressions; however, we observe that for the few expressions with low or zero recovery rate (e.g. Nguyen-7, Nguyen-8, and Nguyen-12), GP sometimes exhibits lower NRMSE. One explanation is that GP is more prone to overfitting the expression to the dataset. As an evolutionary approach, GP directly modifies the previous generation's expressions, allowing it to make small "corrections" that decrease error each generation even if the functional form is far from correct. In contrast, in DSR the RNN "rewrites" each expression from scratch each iteration after learning from a gradient update, making it less prone to overfitting.

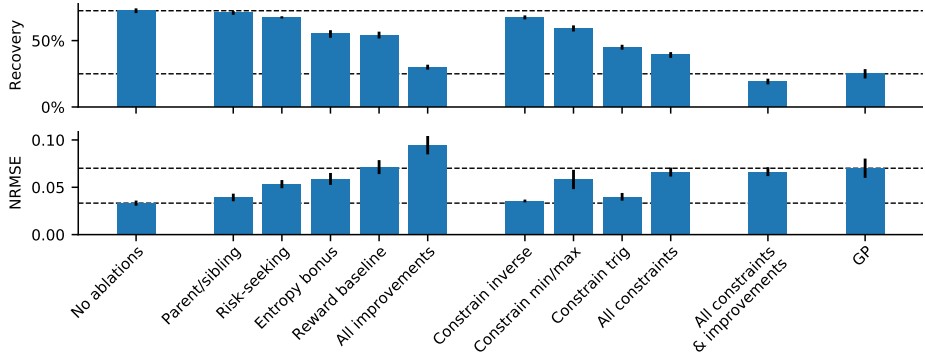

Figure 2: Average recovery (top) and NRMSE (bottom) for various ablations across the 12 Nguyen benchmarks. Dotted lines correspond to DSR (no ablations) and GP baselines. Error bars represented standard error ($n = 10$). Additional descriptions for each ablation experiment are provided in Appendix A.

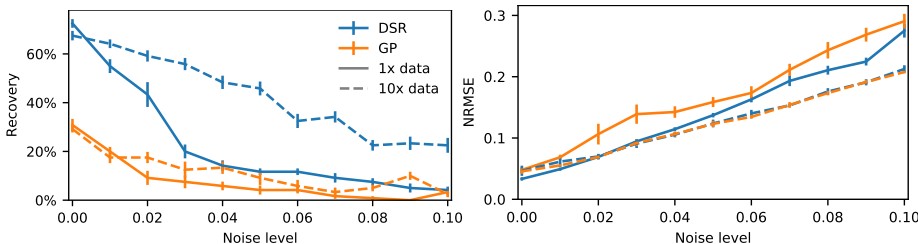

Figure 3: Average recovery (left) and NRMSE (right) for various noise levels across the 12 Nguyen benchmarks. Solid lines represent 20 data points per benchmark (default); dashed lines represent 200 data points per benchmark (10-fold increase). Error bars represent standard error ($n = 10$).

Surprisingly, DSR consistently performed best without a complexity penalty, i.e. $\lambda_{\mathcal{C}} = 0$. Due to the autoregressive nature of the RNN, shorter expressions tend to exhibit higher likelihood than longer ones. We postulate that this property produces a self-regularization effect that precludes the need for an explicit complexity penalty.

**Ablation studies.** Algorithm 3 includes several additional components relative to the "vanilla" Algorithm 2. We performed a series of ablation studies to quantify the effect of each of these components, along with the effects of the various constraints on the search space, and including the parent and sibling as input to the RNN instead of the previous node value. In Figure 2, we performed DSR on the set of 12 Nguyen benchmarks for each ablation. DSR is still competitive with GP even when removing all improvements and all constraints.

**Noisy data and amount of data.** We evaluated the robustness of DSR to noisy data by adding independent Gaussian noise to the dependent variable, with mean zero and standard deviation proportional to the root-mean-square of the dependent variable in the training data. In Figure 3, we varied the proportionality constant from 0 (noiseless) to $10^{-1}$ and compared the performance of GP and DSR across the set of 12 Nguyen benchmarks. DSR still outperforms GP in both recovery rate and NRMSE across noise levels.

Symbolic regression excels in the low-data setting when data is noiseless, hence, the benchmark expressions included herein include only 20 data points (see Table 2). With added noise, increasing the amount of data

smooths the reward function and may help prevent overfitting. Thus, we repeated the noise experiments using the same benchmarks but with 10-fold larger training datasets (200 points data points). As expected, recovery rates tend to increase for both methods; however, DSR maintains a much larger improvement than GP at higher noise levels.

## 5 CONCLUSION

We introduce an unconventional approach to symbolic regression based on reinforcement learning that out-performs a standard GP-based method on recovering exact expressions on benchmark problems, both with and without added noise. Since both DSR and GP generate expression trees, there are many opportunities for hybrid methods, for example including several generations of evolutionary operations in the inner optimization loop. From the perspective of AutoML, the main contributions are defining a flexible distribution over hierarchical, variable-length objects that allows imposing in situ constraints, and using risk-seeking training to optimize best-case performance. Thus, we note that our framework is easily extensible to domains outside symbolic regression, which we save for future work; for example, searching the space of organic molecular structures for high binding affinity to a reference compound. We chose symbolic regression to demonstrate our framework in part because of the large search space, broad applicability, computationally expedient inner optimization loop (sub-second), and availability of vetted benchmark problems and baseline methods.

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

## APPENDIX A

**Hyperparameters.** DSR hyperparameters are listed in Table 3. GP hyperparameters are listed in Table 4. The same hyperparameters were used for all experiments and all benchmark expressions.

**Additional details for performance comparison experiments.** Details of the benchmark symbolic regression problems are shown in Table 2. All benchmarks use the function set $\{+, -, \times, \div, \sin, \cos, \exp, \log\}$. To ensure closures, we use protected versions of operators: $\log$ returns the logarithm of the absolute value of its argument, and $\div$, $\exp$, and $\log$ return 1 for arguments that would cause overflow or other errors. Benchmarks without constants can be recovered exactly, thus recovery is defined by exact correctness (modulo floating point precision error). Note Nguyen-8 can be recovered via $\exp(\frac{x}{x+x}\log(x))$ and Nguyen-11 can be recovered via $\exp(y\log(x))$.

Table 2: Benchmark symbolic regression problem specifications. $U(a, b, n)$ denotes $n$ random points uniformly sampled between $a$ and $b$ for each input variable. Training and testing datasets use different random seeds.

| Name | Variables | Expression | Dataset | Constant? |
|---|---|---|---|---|
| Nguyen-1 | 1 | $x^3 + x^2 + x$ | $U(-1, 1, 20)$ | No |
| Nguyen-2 | 1 | $x^4 + x^3 + x^2 + x$ | $U(-1, 1, 20)$ | No |
| Nguyen-3 | 1 | $x^5 + x^4 + x^3 + x^2 + x$ | $U(-1, 1, 20)$ | No |
| Nguyen-4 | 1 | $x^6 + x^5 + x^4 + x^3 + x^2 + x$ | $U(-1, 1, 20)$ | No |
| Nguyen-5 | 1 | $\sin(x^2)\cos(x) - 1$ | $U(-1, 1, 20)$ | No |
| Nguyen-6 | 1 | $\sin(x) + \sin(x + x^2)$ | $U(-1, 1, 20)$ | No |
| Nguyen-7 | 1 | $\log(x + 1) + \log(x^2 + 1)$ | $U(0, 2, 20)$ | No |
| Nguyen-8 | 1 | $\sqrt{x}$ | $U(0, 4, 20)$ | No |
| Nguyen-9 | 2 | $\sin(x) + \sin(y^2)$ | $U(0, 1, 20)$ | No |
| Nguyen-10 | 2 | $2\sin(x)\cos(y)$ | $U(0, 1, 20)$ | No |
| Nguyen-11 | 2 | $x^y$ | $U(0, 1, 20)$ | No |
| Nguyen-12 | 2 | $x^4 - x^3 + \frac{1}{2}y^2 - y$ | $U(0, 1, 20)$ | No |
| Constant-1 | 1 | $3.39x^3 + 2.12x^2 + 1.78x$ | $U(-1, 1, 20)$ | Yes |
| Constant-2 | 1 | $\sin(x^2)\cos(x) - 0.75$ | $U(-1, 1, 20)$ | Yes |
| Constant-3 | 2 | $\sin(1.5x)\cos(0.5y)$ | $U(0, 1, 20)$ | Yes |
| Constant-4 | 2 | $2.7x^y$ | $U(0, 1, 20)$ | Yes |

Table 3: DSR hyperparameters

| Parameter | Value |
|---|---|
| Batch size | 1,000 |
| Iterations | 1,000 |
| Learning rate ($\alpha$) | 0.0003 |
| Entropy coefficient ($\lambda_{\mathcal{H}}$) | 0.08 |
| Complexity coefficient ($\lambda_{\mathcal{C}}$) | 0 |
| Moving average coefficient ($\beta$) | 0.5 |
| Risk factor ($\epsilon$) | 0.1 |

Table 4: GP hyperparameters

| Parameter | Value |
|---|---|
| Population size | 1,000 |
| Generations | 1,000 |
| Fitness function | NRMSE |
| Initialization method | Full |
| Selection type | Tournament |
| Tournament size ($k$) | 3 |
| Crossover probability | 0.5 |
| Mutation probability | 0.1 |
| Minimum subtree depth ($d_{\min}$) | 0 |
| Maximum subtree depth ($d_{\max}$) | 2 |

For benchmarks with constants, constants are optimized using BFGS with an initial guess of 1.0 for each constant. We ensured that all benchmarks with constants do not get stuck in a poor local optimum when optimizing with BFGS and the candidate functional form is correct. Since floating point constants cannot be recovered exactly, for benchmarks with constants we manually determined correctness of the functional form by inspection. Since constant optimization is a computational bottleneck, we limited each expression to three constants for both DSR and GP experiments.

For GP, the initial population of expressions is generated using the "full" method (Koza, 1992) with depth randomly selected between $d_{\min}$ and $d_{\max}$. The selection operator is defined by deterministic tournament selection, in which the expression with the best fitness among $k$ randomly selected expressions is chosen. The crossover operator is defined by swapping random subtrees between two expressions. The point mutation operator is defined by replacing a random subtree with a new subtree initialized using the "full" method with depth randomly selected between $d_{\min}$ and $d_{\max}$.

**Additional details for ablation studies.** In Figure 2, "Parent/sibling" denotes that the previous node of the traversal is provided as input to the RNN, rather than the parent and sibling node. "Risk-seeking" denotes no risk-seeking, equivalent to $\epsilon = 1$. "Entropy bonus" denotes no entropy bonus, equivalent to $\lambda_{\mathcal{H}} = 0$. "Reward baseline" denotes no reward baseline, equivalent to $\beta = 0$. "All improvements" denotes combining ablations for Parent/sibling, Risk, Entropy, and Baseline. "Constrain trig" denotes no constraint precluding nested trigonometric operators. "Constrain inverse" denotes no constraint precluding inverse unary operators. "Constrain min/max" denotes no constraint precluding minimum or maximum length. (If the maximum length is reached, the expression is appended with $x$ until complete.) "Constraints" denotes combining ablations for Trig, Inverse, and Min/max. "All constraints & improvements" denotes combining all ablations.

**Training curves.** Figures 4 and 5 show the reward $(1/(1 + \text{NRMSE}))$ and recovery rate, respectively, as a function of training step (DSR iteration or GP generation). For benchmarks with constants, the constant optimizer can allow both algorithms to quickly reach reward near 1.0. For these benchmarks, we provide zoomed inset plots demonstrate if and when all independent training runs correctly recovery the expression (Figure 4: Constant-1, Constant-2, and Constant-3). Note that the NRMSE and recovery values in Table 1 correspond to the final point on each curve in Figures 4 and 5, respectively.

**Additional subroutines.** DSR includes several subroutines used when sampling an expression from the RNN and during training. In Subroutine 1, we describe the function ApplyConstraints$(\psi, \mathcal{L}, \tau)$ used in Algorithm 1, which zeros out the probabilities of tokens that would violate any given constraints. Within this subroutine, the user-specific function ViolatesConstraint$(\tau, \mathcal{L}_i)$ returns TRUE if adding the $i$th token of the library $\mathcal{L}$ to the partial traversal $\tau$ would violate any user-specified constraints, and FALSE otherwise.

In Subroutine 2, we describe the function ParentSibling$(\tau)$ used in Algorithm 1, which computes the parent and sibling of the next token to be sampled. This subroutine uses the following logic. If the final node in the partial traversal is a unary or binary operator, then that node is the parent and there is no sibling. Otherwise, the subroutine iterates backward through the traversal until finding a node with an unselected child node. That node is the parent and the subsequent node is the child. Within this subroutine, the function Arity$(\tau_i)$ simply returns the arity (number of arguments) of token $\tau_i$, i.e. two for binary operators, one for unary operators, or zero for input variables and constants.

In Subroutine 3, we describe the function OptimizeConstants$(\tau, X, y)$ used in Algorithms 2 and 3, which optimizes the placeholder constants $c$ of an expression with respect to input dataset $(X, y)$ using a black-box optimizer, e.g. BFGS. Within this subroutine, the function Instantiate$(\tau)$ instantiates the symbolic expression as a function $f(X; c)$ with inputs $X$ and parameters (constants) $c$, and the function ReplaceConstants$(\tau, c^\star)$ replaces the placeholder constants in the expression with the optimized constants.

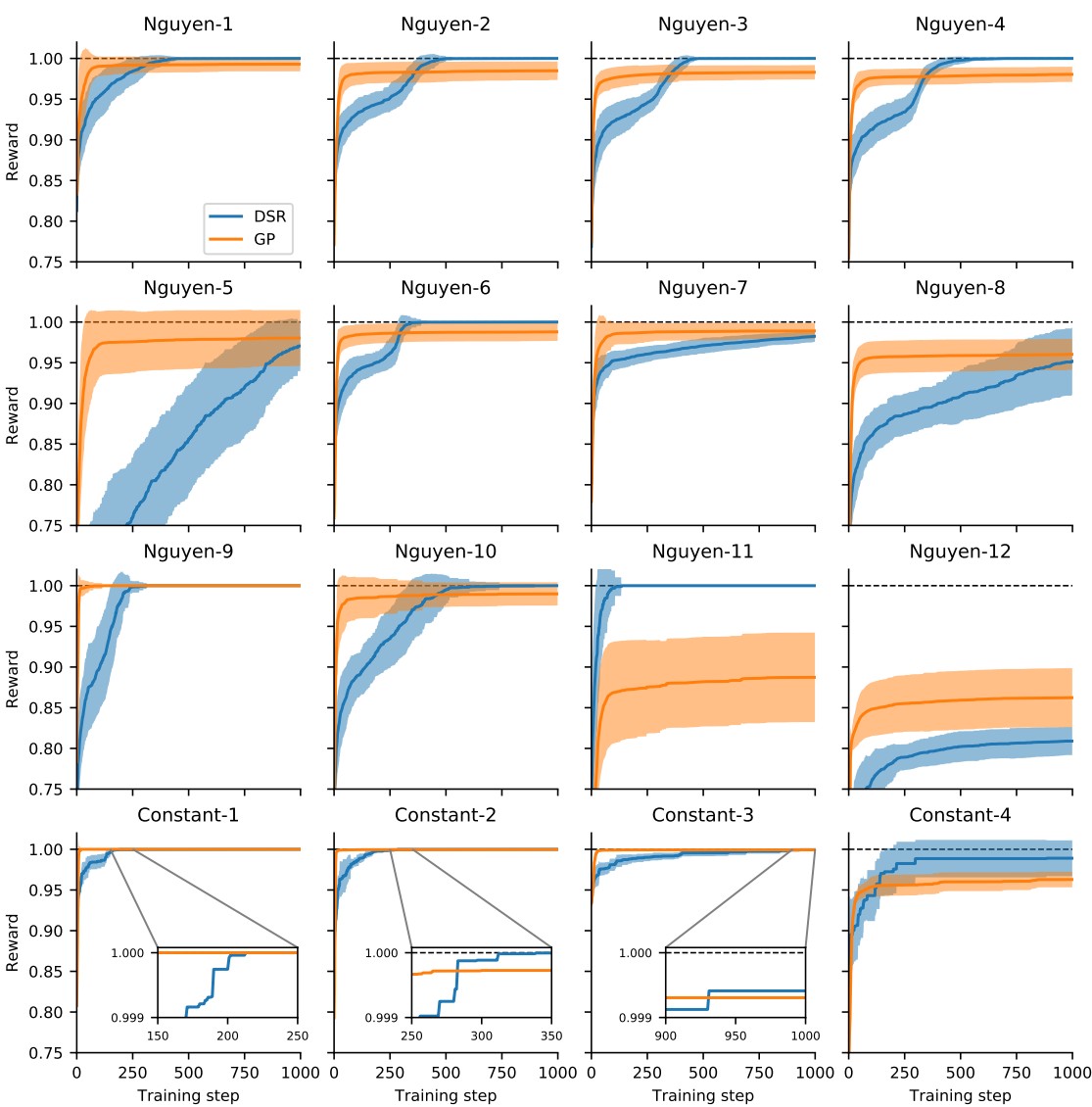

Figure 4: Reward training curves for DSR and GP for each of the 16 benchmark expressions. Each curve shows the best reward $(1/(1 + \text{NRMSE}))$ found so far as a function of training step, averaged across all $n = 100$ independent training runs. A value of 1.0 denotes that all training runs have recovered the correct expression by that training step. Error bands represent standard deviation $(n = 100)$.

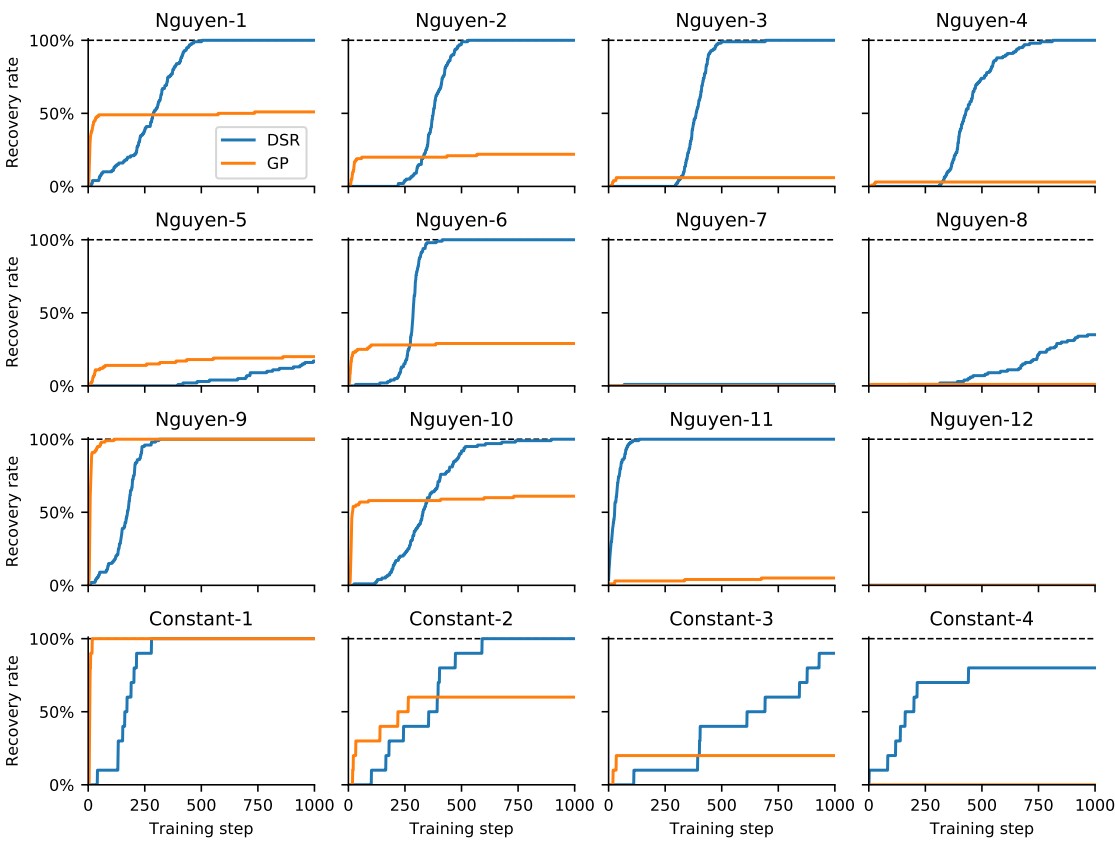

Figure 5: Recovery rate training curves for DSR and GP for each of the 16 benchmark expressions. Each curve shows the fraction of independent training runs that correctly recovered the benchmark expression as a function of training step. A value of 100% denotes that all training runs have recovered the correct expression by that training step.

---

**Subroutine 1:** Apply generic constraints *in situ* when sampling from the RNN

---

1 **function** ApplyConstraints($\psi, \mathcal{L}, \tau$)

    **input** : Categorical probabilities $\psi$; corresponding library of tokens $\mathcal{L}$; partially sampled traversal $\tau$

    **output:** Adjusted categorical probabilities $\psi$

2     $L = |\psi|$                                         `// Length of library`

3     **for** $i = 1, \ldots, L$ **do**

4         **if** ViolatesConstraint($\tau, \mathcal{L}_i$) **then**

5             $\psi_i \leftarrow 0$                           `// Constrain that token`

6     **end**

7     $\psi \leftarrow \frac{\psi}{\sum_i \psi_i}$                               `// Normalize back to 1`

8     **return** $\psi$

---

**Subroutine 2:** Computing parent and sibling inputs to the RNN

---

1 **function** ParentSibling($\tau$)

    **input** : Partially sampled traversal $\tau$

    **output:** Parent and sibling tokens of the next token to be sampled

2     $L = |\tau|$                             `// Length of partial traversal`

3     counter $= 0$                 `// Counter for number of unselected nodes`

4     **if** Arity($\tau_L$) $> 0$ **then**

5         parent $= \tau_L$

6         sibling $=$ empty

7         **return** parent$\|$sibling

8     **for** $i = L, \ldots, 1$ **do**                       `// Iterate backward`

9         counter $\leftarrow$ counter $+$ Arity($\tau_i$) $- 1$

10        **if** counter $= 0$ **then**

11            parent $= \tau_i$

12            sibling $= \tau_{i+1}$

13            **return** parent$\|$sibling

14     **end**

---

**Subroutine 3:** Optimize the constants of an expression (inner optimization loop)

---

1 **function** OptimizeConstants($\tau, X, y$)

    **input** : Expression $\tau$ with placeholder constants $c$; input dataset $(X, y)$

    **output:** Expression $\tau^\star$ with optimized constants $c^\star$

2     $f(X, c) =$ Instantiate($\tau$)         `// Instantiate the symbolic expression`

3     $c^\star = \arg\min_c \|y - f(X, c)\|_2^2$       `// Minimize error (e.g.  with BFGS)`

4     $\tau^\star =$ ReplaceConstants($\tau, c^\star$)       `// Replace placeholder constants`

5     **return** $\tau^\star$

---

