# OpenReview forum: "Deep symbolic regression"
_ICLR.cc/2020/Conference — Reject_

### Official Review · AnonReviewer1 · 2019-10-17
**Official Blind Review #1**

**Rating:** 3

**Review:**

This paper does a good job at specifying a solution, but never states the problem.

For the problem specification, please see the introductory paragraph in https://arxiv.org/pdf/1905.11481.pdf which I quote here to inform other readers:
"In 1601, Johannes Kepler got access to the world’s best data tables on planetary orbits, and after 4 years and about 40 failed attempts to fit the Mars data to various ovoid shapes, he launched a scientific revolution by discovering that Mars’ orbit was an ellipse [1]. This was an example of symbolic regression: discovering a symbolic expression that accurately matches a given data set. More specifically, we are given a table of numbers, whose rows are of the form {x1, ..., xn, y} where
y = f(x1, ..., xn), and our task is to discover the correct symbolic expression for the unknown mystery function f, optionally including the complication of noise."

For people familiar with policy gradients and RNNs, you need only look at the policy RNN in Figure 1. This is a standard approach for sampling a symbolic expression (just as is often done when an RNN composes another net in AutoML). However, note that the authors introduce a bias (input keeps track of parents and siblings) to effectively incorporate hierarchy. Could the authors please expand on their heuristics for automatically choosing the (parent, sibling) input pair? Adding this to Algorithm 1 would help. It would also help with clarity if you could please add the fitting of the parameters of the symbolic expressions using BFGS to the pseudocode.

The RL approach is standard and the authors have executed it carefully and conducted the necessary ablations. However, the axes in Figure 2 should be improved.

The experiments indicate that the proposed RL approach works better than genetic programming (GP) for what appears to be a simple benchmark. However, this is hard to judge. To truly understand the experiments, I advise readers to first look at: https://researchrepository.ucd.ie/bitstream/10197/3528/1/uy_gpem.pdf

I would have loved to see training and test curves, mathematical expressions for the reward so it is unambiguous, ablations of the dataset (eg varying the number of data). I assume a single net with the same parameters applies to all expressions. Is this correct?

While applying policy gradients to symbolic regression is a great idea, the write up of this paper needs to improve substantially.



**Experience Assessment:**

I have published in this field for several years.

**Review Assessment: Checking Correctness Of Derivations And Theory:**

I assessed the sensibility of the derivations and theory.

**Review Assessment: Checking Correctness Of Experiments:**

I assessed the sensibility of the experiments.

**Review Assessment: Thoroughness In Paper Reading:**

I read the paper at least twice and used my best judgement in assessing the paper.

---

> ### Author Response · Authors · 2019-11-12
> **We thank the reviewer for many useful suggestions**
>
> Please see our revised version with the requested additions and improvements.
>
> Thank you for pointing out the missing problem definition---we apologize for this oversight. We added a formal definition of the symbolic regression problem to the first paragraph of the Introduction.
>
> We added Algorithm 1 (new), which describes in pseudocode the process of sampling an expression from the RNN. (Before, this sampling process was only described in the body text and via the Figure 1 illustration.) This algorithm includes computing the next parent and sibling node given a partial pre-order traversal. As requested, we added pseudocode and a written description of the logic for this subroutine in Subroutine 2.
>
> We updated the main DSR Algorithms 2 and 3 (previously numbered 1 and 2) to explicitly include a step for constant optimization. Further, we added Subroutine 3, which provides additional details for this step.
>
> We added the mathematical expression for NRMSE, and improved the axes and labels in Figure 2---thank you for pointing this out. We added Figure 4 in Appendix A, which shows training curves for DSR and GP for each benchmark expression, averaged across the $n = 100$ independent training runs. As expected based on the NRMSE columns of Table 1, the DSR curve surpasses GP at some point during training for most benchmarks. We note that test curves look almost identical (with only a very small differences in NRMSE), so we excluded the corresponding test curves for brevity.
>
> We also added Figure 5 in Appendix A, which shows training curves of the recovery rate, as this tells a slightly different story than Figure 4. We see that recovery using GP tends to increase $\textit{faster}$ than DSR but plateaus lower, indicating that it is more sample efficient but less effective at consistently finding the correct expression. In contrast, DSR increases more slowly but tends to have a $\textit{higher plateau}$, as it recovers the correct expression more often.
>
> The number of data points determines the smoothness of the reward function. This is especially important with noisy inputs, where for small datasets one can find expressions that actually fit $\textit{better}$ than the ground truth expression (hence why recovery rate eventually drops to nearly zero in Figure 3). Thus, we conducted the suggested experiments in which we varied the amount of data, and added these results to the noise experiments in Figure 3. As expected, recovery rates increase (for both DSR and GP) for noise $> 0$ when the size of the dataset increases 10-fold (from 20 to 200 points per benchmark), since the learning signal becomes smoother with more data. However, we found this effect to be more pronounced for DSR than for GP; in fact, DSR with the largest amount of noise and 10x data rivals the performance of GP with no noise and 1x data.
>
> We clarify that while the same hyperparameters are used for all experiments, the parameters of the RNN differ for each expression and for each independent training run. That is, each benchmark task is trained independently. For interested readers, we expound upon this subtlety by noting that various works using RNN-based policy gradient approaches can be categorized by considering what is the end product of training. In some works, the end product is the RNN itself. In these cases, the RNN is a "solver," which maps an input space to a solution for a particular problem type. For example, Bello et al. [1] train an RNN solver for the traveling salesman problem (TSP), in which the input to the RNN is a sequence of coordinates and the output is an ordering of those coordinates. This single trained RNN solves many instances of TSP. In other settings (including ours), the end product is the best sample from the RNN seen during training for a particular task. This is the case in Abolafia et al. [2], who apply policy gradients to the problem of program synthesis.  Their data, input/output pairs for a mystery program, are only used in the reward computation, and their algorithm is retrained for each dataset. Similarly, the input in symbolic regression (namely, the dataset of $(X, y)$ pairs) is only used in the computation of the reward function---the data itself is never used as an input to the RNN. Thus, for a particular symbolic regression problem (i.e. particular dataset), the RNN is retrained for that single task, and the end product is the best found expression(s). We clarify this by adding a return statement (returning the best found expression) to the end of the DSR algorithm pseudocode (Algorithms 2 and 3).
>
> Lastly, we improved clarity in describing how we conducted experiments (see first paragraph of Results).
>
> Thank you again for your time and insights. Please let us know if there are any additional ways we can improve clarity.
>
> [1] Bello et al., Neural combinatorial optimization with reinforcement learning, 2017
>
> [2] Abolafia et al., Neural program synthesis with priority queue training, 2018

---

### Official Review · AnonReviewer3 · 2019-10-23
**Official Blind Review #3**

**Rating:** 6

**Review:**

This paper presents deep symbolic regression (DSR), which uses a recurrent neural network to learn a distribution over mathematical expressions and uses policy gradient to train the RNN for generating desired expressions given a set of points. The RNN model is used to sample expressions from the learned distribution, which are then instantiated into corresponding trees and evaluated on a dataset. The fitness on the dataset is used as the reward to train the RNN using policy gradient. In comparison to GP, the presented DSR approach recovers exact symbolic expressions in majority of the benchmarks.

Overall, this paper presents a novel technique of using an RNN to learn a distribution over mathematical expressions, and using the fitness of sampled expressions as the reward signal to train the RNN using policy gradient. The idea of using the parent and sibling nodes to predict expression nodes in an autoregressive fashion is also interesting, which exploits the fact that the operators are either binary or unary. The experimental results show that it outperforms genetic programming as well as the ablation study shows the usefulness of different extensions.

Grammar VAE (Kusner et al. 2017) learns a distribution over parse trees in a grammar and then uses Bayesian optimization to search over this space to perform symbolic regression. It would be important to empirically evaluate how GVAE performs on these symbolic regression tasks compared to DSR.

During the search for expressions using DSR, it wasn’t clear why the algorithm chose all constants to be 1 for the first 12 benchmarks. Is it because the RNN never chose the constants in the learnt distribution or the BFGS algorithm prefers constants to be 1? Also, could it be the case that GP is being unfairly penalized for such cases as it might be trying to learn real-valued constants. Would it be possible to report what expressions GP came up with for the first 12 benchmarks?

How was the Recovery metric calculated? Does it require exact syntactic match or it also allows for semantically equivalent expression (that might be different syntactically)?

What are the runtimes for the REINFORCE and GP methods? It wasn’t clear how big the total search space of expressions was without constants. How would a random search that enumerates over all expressions and uses BFGS to compute constants work?

The evaluation is only performed on 16 expressions and only 4 expressions have real-valued constants. It would be good to evaluate the technique on more benchmarks especially with real-valued constants, and larger expressions. Is Nguyen-4 already getting to maximum length of 30? Can the expressions from AI Feyman (Udrescu & Tegmark 2019) or synthetically generated expression be used?

From the experiment benchmarks, it seems only 4 expressions had real-valued constants, and for these benchmarks GP did quite well in terms of NRMSE. What expression is GP coming up with for these benchmarks?

Matt J. Kusner, Brooks Paige, and José Miguel Hernández-Lobato. 2017. Grammar variational autoencoder. ICML 2017

**Experience Assessment:**

I have published one or two papers in this area.

**Review Assessment: Checking Correctness Of Derivations And Theory:**

I carefully checked the derivations and theory.

**Review Assessment: Checking Correctness Of Experiments:**

I carefully checked the experiments.

**Review Assessment: Thoroughness In Paper Reading:**

I read the paper thoroughly.

---

> ### Author Response · Authors · 2019-11-07
> **We thank the reviewer for the insightful comments and feedback**
>
> We evaluated DSR on the benchmark expression showcased by GrammarVAE: $1/3 + x + \sin(x^2)$. We reproduced their experimental setup by using an identical input dataset and library of tokens. Applying DSR, we recovered the exact expression in 5 out of 100 attempts (recovery rate: 5%) with an average NRMSE of $0.049 \pm 0.017$. In contrast, GrammarVAE never recovers their own benchmark expression.
>
> Regarding choosing constants with value 1, we clarify that the first 12 benchmarks (Nguyen-1 through Nguyen-12)---as defined in Uy et al. [1]---do not include the constant token as part of the library. (See Table 2 and first sentence of Results.) Thus, neither DSR nor GP chose constants for these benchmarks. We hope this allays any concerns that GP was unfairly penalized. Further, we emphasize that our implementation of GP $\textit{did}$ optimize constants using an identical constant optimizer (BFGS) for the benchmarks with constants (Constant-1 through Constant-4), so GP was not penalized for those experiments either. We agree that it would have been an unfair comparison if GP could only guess at constants while DSR could optimize them. Thus, for experiments performed herein, constant optimization using BFGS is essentially subsumed as part of the computation of the reward, i.e. $R(\tau)$ can be viewed as $R(\textrm{BFGS}(\tau))$, where $\textrm{BFGS}(\tau)$ is the symbolic expression whose constants have been optimized using BFGS.
>
> The recovery metric requires exact $\textit{semantic}$ correctness. For example, $a + b$ is equivalent to $b + a$, and $\sin^2(x) + \cos^2(x)$ is equivalent to $1$. The Appendix provides additional details on how recovery is confirmed using a computer algebra system (SymPy).
>
> For Nguyen-1 through Nguyen-12, average runtimes on a single core are 27.5 $\pm$ 19.2 min for DSR and 6.0 $\pm$ 1.0 min for GP. Experiments with constants are substantially more expensive, averaging roughly $10 - 12$ hr for either method; clearly, this is largely dominated by the inner optimization loop (BFGS), which is identical for GP and DSR.
>
> While the exact size of the search space is non-trivial to compute [2], an upper bound can be estimated as $|\mathcal{L}|^T$, where $|\mathcal{L}|$ is the size of the library (between 9 and 11, depending on the number of input variables and whether constant was part of the library), and $T$ is the maximum length (30), resulting in between $10^{28}$ and $10^{31}$ possible expressions. For both DSR and GP, we evaluate at most $10^6$ expressions. Simple random search using BFGS could be conducted as a separate baseline---and may even perform well in terms of NRMSE---but we would not expect it to recover exact expressions for most benchmarks.
>
> Regarding including more benchmarks with constants, we actually contend that the inclusion of constants is not particularly effective in differentiating the performance of DSR versus GP, compared to using benchmarks without constants. To support this claim, we make several observations: 1) The computational bottleneck is constant optimization, as shown above. 2) Both DSR and GP use the same constant optimizer. 3) Constant optimization can be thought of as subsumed by the black-box reward function, as discussed above. Taken together, we believe that benchmarks with constants do not make the symbolic search itself particularly more challenging than, say, replacing the constant with another input variable. Further, by focusing on experiments without constants, the reduced computational cost allowed us to conduct sufficient independent runs to perform statistical significance tests between GP and DSR performance.
>
> We are wary of evaluating our approach on synthetically generated expressions, as these are highly biased by their synthesis technique. Rather, we rely on established benchmark expressions that were specifically designed for the symbolic regression task and have been vetted by the symbolic regression community [3]. AI Feynman---an excellent addition to the field, we might add---is at its heart a recursive simplification algorithm, which in its innermost step requires a searching algorithm (in their case, brute force search). Therefore, any symbolic regression algorithm can be incorporated into this innermost step. Here we focus on the search itself rather than simplifying the problem; however, a hybrid approach could be extremely effective for real-world applications, which we leave for future work.
>
> Solutions to Nguyen-4 sometimes reach the maximum length of 30, but most solutions involve significant polynomial factoring, which can substantially reduce the size of the tree.
>
> We will provide the requested GP expressions in a separate post.
>
> [1] Uy et al., Semantically-based crossover in genetic programming, 2011.
>
> [2] Ebner, On the search space of genetic programming and its relation to nature's search space, 1999.
>
> [3] White et al., Better GP benchmarks: community survey results and proposals, 2013.

---

> ### Author Response · Authors · 2019-11-07
> **As requested, we provide the best failed GP expressions for all benchmarks**
>
> As requested, below we provide the best incorrect expressions found by the GP for all 16 benchmark expressions.
>
> $\textrm{Nguyen-1: } \sin(x) + x e^{x-\sin(\sin(\cos(x)))}   e^{x - \sin(\cos(\cos(\log(x) + \sin(x^2))))}$
>
> $\textrm{Nguyen-2: } \frac{x^2 e^x}{2 + x\cos(x)}$
>
> $\textrm{Nguyen-3: } \frac{x e^x}{\cos\left( \frac{x}{\cos\left(\frac{x^2-x}{\cos(\sin(\cos(x)))}\right) \cos\left(\frac{\sin(x)-x}{\log(2x)}\right)} \right)}$
>
> $\textrm{Nguyen-4: } xe^{2x}\cos(\cos(\cos(\cos(\cos(\cos(x + e^x + \sin(x + \sin(\sin(2x)) - \log(\sin(x)))))))))$
>
> $\textrm{Nguyen-5: } -\cos(x^2\log(x)-x)$
>
> $\textrm{Nguyen-6: } \sin(\sin(x))+\sin(\sin(\sin(x))) + \sin(\sin(\sin(\sin(\sin(x)+\frac{x^2}{x+\cos(2x)}))))\sin(\sin(x))$
>
> $\textrm{Nguyen-7: } \sin(x)\left( x + \sin\left( \frac{\frac{x}{\sin(x)+x}}{\sin\left(\sin\left(\frac{\frac{\sin(x^2)}{\sin(x\sin(x))}+x}{x}\right)\right)} \right) \right)$
>
> $\textrm{Nguyen-8: } \log\left( \cos(x)+2x+\frac{x}{\cos(x)+x+\cos(\log(x))+\frac{x}{\cos(\log(\log(x))) + \sin(\sin(\sin(x)))}} \right)$
>
> $\textrm{Nguyen-9: N/A (GP always recovered the exact expression)}$
>
> $\textrm{Nguyen-10: } 2x + \frac{\sin\left(\sin\left( y\sin\left( e^y\sin\left( xe^{\sin\left( \sin(x)e^{\sin\left( \frac{e^x}{\sin(y)} \right)} \right)} \right) \right) \right)\right)}{e^y}$
>
> $\textrm{Nguyen-11: } \cos\left( \frac{y}{x} - y\sin\left(   \cos(\cos(y)) \left(\frac{y}{x} - \sin\left( \frac{y}{x} - \cos\left(\cos\left(\frac{y^2}{x}\right)\right) \right) \right) \right)\right)$
>
> $\textrm{Nguyen-12: } -y\cos(y)$
>
> $\textrm{Constant-1: N/A (GP always recovered the exact expression)}$
>
> $\textrm{Constant-2: } 1.848x\left( x + x\cos\left(\frac{2.041x}{\sin(x)}\right) \right) -0.750$
>
> $\textrm{Constant-3: } \sin(x(1.098+0.401(\cos(y)+x^2\sin(\sin(\sin(x))(\sin(\sin(x))-x(0.685+y))))))$
>
> $\textrm{Constant-4: } \log(\log(\sin(x)) - 15.623) \left( \cos(1.591-y+xy) + \cos(x-\sin(\sin(\log(\sin(\sin(\sin(x))))-4.337))) \right)$
>
> We note that while results such as these are certainly very interesting---especially for real-world applications---we ultimately felt that including such post-mortem analysis within the paper was outside the scope of this methodology work.

---

### Official Review · AnonReviewer2 · 2019-10-25
**Official Blind Review #2**

**Rating:** 1

**Review:**

This paper presents a RNN-RL based method for the symbolic regression problem. The problem is new (to Deep RL) and interesting. My main concern is about the proposed method, where the three RL related equations (not numbered) at page 5 are also direct copy-from-textbook policy gradient equations without specific adaptation to the new application considered in this paper, which is very strange. The two conditional probability definitions considered at page 3 are not mentioned in later text. These are only fractions of the underlying method and by reading the paper back and forth several times, it is not clear of the basic algorithmic flowchart, let alone more detailed description of the related parameters. Without these information, it is impossible to have a fair judge of the novelty and feasibility of the proposed method. The empirical results are also limited in small dataset, which makes it hard to verify the generality of the superior claim.


**Experience Assessment:**

I have read many papers in this area.

**Review Assessment: Checking Correctness Of Derivations And Theory:**

I assessed the sensibility of the derivations and theory.

**Review Assessment: Checking Correctness Of Experiments:**

I assessed the sensibility of the experiments.

**Review Assessment: Thoroughness In Paper Reading:**

I read the paper thoroughly.

---

> ### Author Response · Authors · 2019-11-15
> **Thank you for reviewing our work; we provide responses below**
>
> Regarding writing the policy gradient equations "without specific adaptation to the new application considered in this paper," the benefit of the REINFORCE rule is that it is very general: it applies to all cases in which one has a distribution over sequences (for which the likelihood is differentiable) and a black-box reward function; there is no need for "specific adaptation" of this theorem. Rather, one must simply define the relevant variables with respect to the application: namely, the sequence $\tau$, the distribution $p(\tau | \theta),$ and the black-box reward $R(\tau).$ We clearly define each of these in the text: $\tau$ is the pre-order traversal of an expression sampled from the RNN (Section 3.1). The distribution $p(\tau | \theta)$ and its likelihood computation are defined in their own section (Section 3.1). The black-box reward $R(\tau)$, based on the mean-squared error between the candidate expression and benchmark dataset, is also described in its own section (Section 3.2).
>
> These equations are critical background information which we believe are necessary to include in any RNN-based policy gradient paper that optimizes discrete objects. In fact, we are not aware of a single such paper that does not include these equations in some form [1 - 7].
>
> Regarding the conditional probability equation "not mentioned in later text," we kindly point out that this result---which describes how to compute the joint likelihood of an expression under the RNN, $p(\tau | \theta)$, as a product of conditionals---is used 7 additional times throughout the paper, notably in the gradient computation in Algorithm 2 and 3. Further, this statistics result is provided for background only---in fact, many other RNN-based policy gradient papers elect to omit this detail [6, 7]; however, we included it for thoroughness.
>
> We feel that these comments focus mostly on background material (which we have found to be standard in the literature [1 - 7]) rather than our contributions. The contributions of our method are not in simply applying the REINFORCE rule to optimize expressions, but rather 1) defining a distribution $p(\tau | \theta)$ over $\textit{hierarchical}$ objects (i.e, expressions) by providing the parent and sibling as input to the RNN, 2) incorporating prior constraints in situ (i.e., when sampling the expression), and 3) developing a risk-seeking strategy to focus efforts on improving the best expressions as opposed to the average expression. We implore the reviewer to consider and assess these contributions in their review.
>
> We value clarity above all else. To this end, we have added Algorithm 1 (the previous Algorithms 1 and 2 are now numbered 2 and 3), which provides pseudocode for sampling an expression from the RNN. Thus, the sampling process is now described in three separate ways: with pseudocode (Algorithm 1), with a textual description (Section 3.1), and with an illustration and its supporting caption (Figure 1). Further, we have added pseudocode and additional textual descriptions in the Appendix for several additional subroutines used during sampling and training: applying in situ constraints (Subroutine 1), computing the parent and sibling inputs to the RNN (Subroutine 2), and optimizing the constants of an expression (Subroutine 3). Regarding "more detailed descriptions of the related parameters," we believe we have provided full descriptions of all variables and parameters---we kindly ask the reviewer to point out instances in which we can increase the clarity of describing parameters.
>
> Regarding the size of the experiments, the benchmark expressions used herein have been developed specifically for symbolic regression and have been vetted by the symbolic regression community [8]. Thus, we focused our computational efforts on increasing statistical power ($n = 100$ independent training runs for each benchmark) across a set of vetted benchmarks, rather than a longer list of benchmarks.
>
> [1] Zoph and Le, Neural architecture search with reinforcement learning, 2016
>
> [2] Ramachandran et al., Searching for activation functions, 2017
>
> [3] Bello et al., Neural optimizer search with reinforcement learning, 2017
>
> [4] Bello et al., Neural combinatorial optimization with reinforcement learning, 2017
>
> [5] Popova et al., MolecularRNN: Generating realistic molecular graphs with optimized properties, 2019
>
> [6] Abolafia et al., Neural program synthesis with priority queue training, 2018
>
> [7] Liang et al., Memory augmented policy optimization for program synthesis and semantic parsing, 2018
>
> [8] White et al., Better GP benchmarks: community survey results and proposals, 2013

---

### Decision · Program_Chairs · 2019-12-19

**Decision:**

Reject

**Comment:**

This paper suggests using RNN and policy gradient methods for improving symbolic regression. The reviewers could not reach a consensus, and due to concerns about the clarity of the paper and the extensiveness of the experimental results, the paper does not appear to currently meet the level of publication.

Also, while not mentioned in the reviews, there appears to be some work on symbolic regression aided by deep learning, (see for example, https://twhughes.github.io/pdfs/cs221_final.pdf, which was found by searching "symbolic regression deep learning")---I would thus also recommend the authors do a more thorough literature search for future revisions.